# Unsupervised Domain Adaptation with Raman Spectroscopy for Rapid Autoimmune Disease Diagnosis

**DOI:** 10.3390/s25196186

**Published:** 2025-10-06

**Authors:** Ziyang Zhang, Yang Liu, Cheng Chen, Xiaoyi Lv, Chen Chen

**Affiliations:** 1College of Software, Xinjiang University, Urumqi 830046, China; 107552305028@stu.xju.edu.cn (Z.Z.); gumpliu404@163.com (Y.L.); chenchengoptics@gmail.com (C.C.); xjuwawj01@163.com (X.L.); 2Key Laboratory of Signal Detection and Processing, Xinjiang University, Urumqi 830046, China; 3Xinjiang Cloud Computing Application Laboratory, Xinjiang University, Karamay 834099, China; 4Hongyou Software Co., Ltd., Karamay 834099, China

**Keywords:** unsupervised domain adaptation (UDA), conditional domain adaptation, pseudo label, autoimmune disease, computer-aided diagnosis (CAD)

## Abstract

Autoimmune diseases constitute a broadly prevalent category of disorders. Conventional computer-aided diagnostic (CAD) techniques rely on large volumes of data paired with reliable annotations. However, the diverse symptomatology and diagnostic complexity of autoimmune diseases result in a scarcity of reliably labeled biological samples. In this study, we propose a pseudo-label-based conditional domain adversarial network (CDAN-PL) framework by integrating Raman spectroscopy with domain adaptation technology, enabling label-free unsupervised transfer diagnosis of diseases. Compared to traditional unsupervised domain adaptation techniques, our CDAN-PL framework generates reliable pseudo-labels to ensure the robust implementation of conditional adversarial methods. Additionally, its spectral data-adaptive feature extraction techniques further solidify the model’s superiority in Raman spectroscopy-based disease diagnosis. CDAN-PL exhibits excellent performance in homologous transfer tasks, achieving an average accuracy of 92.3%—surpassing the baseline models’ 80.81% and 86.4%. Moreover, it attains an average accuracy of 90.05% in non-homologous transfer tasks, further validating its generalization capability.

## 1. Introduction

Autoimmune diseases refer to a series of disorders caused by the immune system mistakenly attacking the body’s own tissues and organs, characterized by a wide variety of types, involvement of systemic symptoms, and complex pathogenesis. The complexity and diversity of autoimmune diseases is manifested in that the etiology of a single autoimmune disease is reflected by a combination of symptoms; correspondingly, a single symptom may imply the involvement of pathological processes of multiple autoimmune diseases [1]. Consequently, their diversity and complexity pose significant challenges for clinicians in clinical diagnosis. Typical autoimmune diseases include ankylosing spondylitis (AS), osteoarthritis (OA), and Sjögren’s syndrome (SS). Ankylosing spondylitis (AS), also known as axial spondyloarthritis, is a chronic inflammatory disease primarily affecting the axial skeleton [2]. The pathogenesis of AS has not been fully elucidated; it is generally believed to involve interactions among genetic, immune, and environmental factors, including abnormal expression of the HLA-B27 antigen, immune system dysregulation, and environmental factors [3]. Osteoarthritis (OA) is a whole-joint disease involving structural changes in articular cartilage, subchondral bone, ligaments, joint capsules, synovium, and periarticular muscles, with pain as its primary symptom [4]. The pathogenesis of OA is mainly attributed to an imbalance between pro-inflammatory and anti-inflammatory mediators, which further leads to low-grade inflammation, cartilage degradation, bone remodeling, and synovial hyperplasia [5]. Sjögren’s syndrome (SS) is a systemic chronic autoimmune disease characterized by immune-mediated damage to the salivary and lacrimal glands, resulting in xerostomia (dry mouth) and xerophthalmia (dry eyes) [6]. SS is generally considered to be caused by factors such as congenital immune barrier dysfunction, infections by pathogens like viruses, and genetic factors.

Clinical diagnostic approaches for the aforementioned diseases typically include clinical symptom assessment, antigen detection, autoantibody testing, biopsy, and imaging examinations [7]. However, conventional diagnostic methods have several limitations: clinical symptom assessment relies on the judgment of experienced physicians; biopsy is often highly invasive; autoantibody testing and imaging examinations are relatively costly; and other diseases that may cause similar symptoms need to be excluded during the diagnostic process before a diagnosis can be confirmed. Raman spectroscopy is a non-invasive analytical technique that enables non-destructive detection, providing molecular fingerprints of samples and quantitative information about their chemical composition. It possesses multiple properties beneficial for medical diagnosis, including high chemical specificity [8], and thus holds great potential in disease diagnosis. With the advancement of artificial intelligence, the integration of spectroscopy with deep learning and machine learning has effectively addressed the limitations of traditional diagnostic methods for autoimmune diseases, allowing Raman spectroscopy to further exert a significant role in the biomedical field, particularly in disease diagnosis [9,10,11,12,13,14]. Leng et al. [15] utilized Raman and Fourier-transform infrared (FTIR) spectra from a total of 119 patients with different types of cancer, performing low-level fusion and feature fusion on the spectra. By employing classifiers such as support vector machine (SVM) and convolutional neural network–long short-term memory (CNN–LSTM), they improved the accuracy of the fusion model by approximately 10%. Yong et al. [16] classified Raman spectra collected from the superficial and deep layers of cartilage in 45 patients with osteoarthritis and 19 patients with osteoporosis (serving as healthy controls). Using a multi-convolutional neural network with sixfold cross-validation, they achieved higher classification accuracy. Yang et al. [17] extracted multi-scale features from surface-enhanced Raman spectroscopy (SERS) data via wavelet transform, and constructed a rapid detection method for liver cancer samples by combining data augmentation and deep learning techniques, achieving an accuracy of 99.38%. Cao et al. [18] designed a one-dimensional residual convolutional neural network (1D-ResNet) architecture to classify tumor tissues of colorectal cancer and visualized and interpreted the fingerprint peaks identified by the deep learning model; their study achieved an accuracy of 98.5% in colorectal cancer detection.

However, existing studies often assume that model training is conducted under the condition that all data are labeled, i.e., supervised learning. In practice, annotating medical data is labor-intensive, resource-consuming, and costly. Furthermore, considering patient privacy, techniques such as de-identification and data desensitization may be required for data processing. Additionally, annotations typically rely on physicians’ clinical diagnostic outcomes. For autoimmune diseases—characterized by complex pathogenesis, high heterogeneity, and the need for exclusionary diagnosis—the accuracy of labels is, to some extent, questionable. Unsupervised learning, which can be performed without labeled training data, enables the learning of internal data structures and distributions, as well as the discovery of potential relationships within the data. It effectively addresses the issue of insufficient data and annotations, holding significant implications for the practical application of artificial intelligence in clinical diagnosis. Although unsupervised learning has been widely applied in the fields of machine learning and deep learning with increasingly mature techniques, research on autoimmune disease diagnosis methods addressing insufficient data annotation remains relatively limited. The framework proposed in this study aims to fill the gap in this field, provide a solid foundation for subsequent research, and further enhance the diversity and feasibility of this research direction.

Transfer learning is a class of machine learning algorithms that aim to leverage existing knowledge to train or refine models in new domains where data or label resources are scarce, thereby improving model performance. Deep neural network-based transfer learning has already played a significant role in fields such as computer vision, signal processing, bioinformatics, and recommendation systems [19,20,21]. Domain adaptation, a specific application scenario of transfer learning, shares the goal of addressing knowledge transfer between source and target domains with transfer learning. However, domain adaptation focuses on reducing the distribution discrepancy between the source and target domains to enable semi-supervised or unsupervised model training on the target domain. Both transfer learning and domain adaptation have extensive applications and great potential in Raman spectroscopy. Jiaqi Hu et al. [22] performed pre-training using 10,000 Raman spectra of 200 substances from the RRUFF database and evaluated the feature transfer performance of CNN-1D, ResNet-1D, and Inception-1D on collected pesticide Raman spectral data. Chen et al. [23] selected three sets of serum Raman spectral data as the source domain and two sets as the target domain, with data augmentation applied. They trained three deep neural network models—CNN-LSTM, GoogLeNet, and ResNet—on the source domain data for disease diagnosis, transferred the models to the target domain, and further improved model performance by constructing a decision-level fusion model combined with logistic regression. Yu Yao et al. [24] implemented unsupervised multiplex biomolecular detection by conducting domain adaptation on 15 different Raman spectra in suspension array technology (SAT) based on Raman spectral encoding. Existing studies have utilized Raman spectroscopy for unsupervised learning and have made significant contributions across various application domains. However, there remains a lack of systematic research focused specifically on unsupervised diagnosis of diseases—particularly autoimmune diseases—using Raman spectroscopy. As summarized in Table 1, a comparison between previous Raman spectroscopy studies combined with AI and this study highlights our distinct approach.

This study addresses the challenges in Raman spectroscopy-based autoimmune disease diagnosis, including AS, OA, and RA, specifically the difficulties in acquiring spectral data and labels, as well as diagnostic complexities arising from the intricate and diverse etiologies of autoimmune diseases. We propose an unsupervised domain adaptation framework with consensus voting for pseudo-label generation. Specifically, we generate pseudo-labels for the target domain by leveraging votes from source domains with feature distributions similar to the target domain, select high-confidence labeled samples to compute and optimize mc-loss [25] and cross-entropy loss, and update the parameters of the feature extractor accordingly. Furthermore, we drew on the domain discriminator framework in conditional domain adaptation [26], concatenating features and labels from both the source and target domains to condition the training of the adversarial network. Our framework achieved optimal performance in mutual adaptation experiments using serum Raman spectral data from three homologous autoimmune diseases. Additionally, we conducted transfer experiments where Raman spectral data of three common non-homologous cancers were transferred to autoimmune disease data, achieving unsupervised disease diagnosis and further validating the generalization capability of our model.

## 2. Materials and Methods

### 2.1. Sample Collection

Serum samples from patients with autoimmune diseases, cancer, and healthy controls used in this study were obtained from Xinjiang Uygur Autonomous Region People’s Hospital, with all studies approved by the ethics committee and written informed consent obtained from all participants. All samples were collected in the morning, and volunteers fasted for 12 h to avoid interference from diet and other factors affecting serum components. All serum samples were derived from fresh blood samples, ensuring the absence of any anticoagulants. For each sample, 3 mL of fresh blood was collected; subsequently, the blood samples were centrifuged at 4000 r/min at 4 °C, and the uppermost transparent liquid was extracted to obtain serum. The serum was aliquoted into centrifuge tubes and stored in a −80 °C refrigerator. Detailed sample collection information for dataset construction is presented in Table 2.

### 2.2. Data Acquisition

For patients with autoimmune disease and their healthy control samples, 10 µL of serum sample was transferred onto a tinfoil-lined slide using a pipette gun, and the serum sample was allowed to dry naturally for approximately 13 min, but not completely. Spectra were collected using a high-resolution confocal Raman spectrometer (LabRAM HR hEvolution, Gora Raman Spectroscopy, Ideal Optics, Shanghai, China) before the sample was fully dried. The excitation wavelength was 785 nm, the excitation time was 15 s, the laser power was 160 mW, and the diameter of the laser spot was 2.2 µm. In this setting, the Raman spectra of the serum samples were measured in the range of 500–2000 cm^−1^. Five locations were randomly selected for each serum sample, and five different spectral signals were recorded.

Spectroscopic data acquisition for patients with cancer and their healthy control samples was performed in practice using different models of confocal spectrometers as well as operational procedures. Raman spectra of serum samples were collected using a confocal Raman spectrometer (LabRAM HR Evolution Raman Spectrometer, Horiba Scientific, Ltd., Beijing, China). The laser wavelength was 532 nm, the laser power was 20 mW, the integration time was 25 s, and the number of integrations was 3. The laser beam was focused through a 10× objective lens into a sterile glass capillary for measurement. The spectral measurement range was 500 to 2000 cm^−1^. Each serum sample was measured three times from different positions.

### 2.3. Data Preprocessing

In the actual acquisition of Raman spectral data, Raman signals collected by spectrometers often contain various types of noise, background radiation, and superimposed random errors. Removing these signals—which may interfere with Raman signals—through a series of spectral preprocessing steps facilitates a clear interpretation of the relationship between Raman spectral lines and the intrinsic biochemical components of samples while better adapting to the requirements of deep learning algorithms. Common Raman spectral preprocessing steps include smoothing for denoising, baseline correction, and normalization.

Baseline correction was performed using the least-squares polynomial fitting method [27], with the polynomial degree set to 3, the number of iterations to 100, and the polynomial loss gradient to 0.001. Mean smoothing was applied for denoising [28], with a smoothing window width of 9. Finally, min–max normalization was conducted to align with the data range required by deep neural networks. Considering the differences between the source and target domains, healthy control samples from both domains were split in a 1:1 ratio to form the source domain dataset and target domain dataset, respectively, with each dataset further divided into training and testing sets in a 7:3 ratio.

To address the issues of small sample size and class imbalance, we further performed data augmentation on the training set. Synthetic Minority Oversampling Technique (SMOTE) [29] is a commonly used algorithm for data balancing and augmentation. This algorithm generates additional samples for specified classes through random interpolation in the feature space, a method that avoids simple duplication of existing samples (ensuring a degree of randomness) while preserving the fundamental characteristics of the original class. SMOTE can mitigate overfitting risks associated with random oversampling algorithms and reduce introduced noise, enabling the model to achieve better performance. During training, we applied the SMOTE algorithm to all classes in both the source and target domain training sets for data augmentation, ensuring each class contained 300 samples.

Additionally, due to differences in the resolution of confocal spectrometers used for acquiring Raman spectral data of cancers and autoimmune diseases, the data dimensions are inconsistent. Prior to conducting the cancer-to-autoimmune disease transfer diagnosis task, we performed principal component analysis (PCA) [30] on both datasets, retaining the first 50 principal components and ensuring their cumulative contribution rate exceeded 99%. The dataset categories and data information involved in the experiments of this study are shown in Table 3.

### 2.4. Proposed Method

For the unsupervised domain adaptation problem addressed in this study, the source domain is defined as $D_{s}={\left\{\left(x_{s},y_{s}\right)\right\}}^{N_{s}}$ and the target domain as $D_{t}={\left\{\left(x_{t}\right)\right\}}^{N_{t}}$, where $x_{s}$ and $y_{s}$ denote the source domain data and corresponding labels, respectively, while $x_{t}$ denotes the target domain data. Due to the requirements of unsupervised domain adaptation (UDA), the target domain labels $y_{t}$ are not included in the target domain dataset, and *N* denotes the number of classes. The source and target domains follow distinct joint distributions, i.e., $P_{s}=\left(x_{s},y_{s}\right)$ and $P_{t}=\left(x_{t},y_{t}\right)$, with $P_{s}\neq P_{t}$. Such discrepancy gives rise to domain shift, and domain adaptation techniques can mitigate this shift, making the distribution of the target domain as similar as possible to that of the source domain, as illustrated in Figure 1. In this study, we constructed the deep model CDAN-PL to reduce domain shift between the source and target domains; the overall framework is presented in Figure 2.

#### 2.4.1. Predictive Labeling and Pseudo-Labeling

We generated pseudo-labels and predicted labels for the target domain through three stages.

(1) Neuron Predictions

The feature extractor *G* is used to extract features from the source domain and target domain data. A dual-branch feature extraction based on a transformer encoder and multi-scale convolution and fusion module is employed to represent and fuse features, and the specific structure of *G* is described in Figure 2a. The fully connected (FC) layer outputs binary classification predictions ${\tilde{y}}_{s}$ or ${\tilde{y}}_{t}$, and this network serves as the backbone. During actual training, source domain and target domain data are input separately, generating corresponding features and predictions. The features extracted by the encoder and the ground-truth labels of the source domain participate in subsequent computations, which will be described in the conditional domain.

(2) Pseudo-label Generation via K-Nearest Neighbors

Given the feature distribution space and predicted labels of source domain data output by the backbone network, we can assign pseudo-labels to target domain features using the k-nearest neighbors (KNN) algorithm. Specifically, KNN predicts the class of a test sample based on the majority class among its K nearest neighbors: if a certain class appears most frequently among the K neighbors, the test sample is predicted to belong to that class. In our approach, we perform KNN classification by projecting the unlabeled target domain feature distribution into the source domain feature space with ground-truth labels. This maximizes the correspondence between pseudo-labels in the target domain and their counterparts in the source domain after feature alignment. Under such circumstances, the pseudo-label generation process can be expressed as:(1)$${\tilde{y}}_{t}=KNN\left(G\left(x_{t}\right),G\left(x_{s}\right),{\tilde{y}}_{s}\right)$$

Pseudo-labels participate in the calculation of cross-entropy loss and influence the feature extractor (G) through backpropagation:(2)$$L_{CE}\left(\theta_{G}\right)=-E_{\left(x_{s},y_{s}\right)∼P_{s}}\left[{\tilde{y}}_{t}\log\left(FC\left(G\left(x_{s}\right)\right)\right)\right]$$

Although the pseudo-label generation strategy we proposed achieves feature-level alignment between the source and target domains, the generated labels inherently contain a certain degree of noise. A large number of incorrectly generated labels may reduce diagnostic accuracy [31,32]. To ensure high reliability of ${\tilde{y}}_{t}$, a confidence threshold κ is used to filter target samples. The filtering rule is as follows:(3)$$\left\{x\in x_{t}|max\left(C\left(x\right)\right)>\kappa \right\}$$

(3) Predictive Label Generation via Decision Fusion

In deep learning, decision fusion is a technique that combines prediction results from multiple models or methods to make final decisions, typically used to improve result accuracy and model robustness. The two pseudo-label generation methods mentioned above each have their own focuses: the neuron prediction method is an end-to-end approach, where the model is treated as a whole, and we only concern ourselves with the predicted class of input data to ensure consistency between pseudo-labels and ground truth labels; the KNN method, on the other hand, focuses on the consistency of the spatial distribution between source and target domain features after alignment, ensuring that features corresponding to pseudo-labels remain consistent with those of the source domain. In domain adversarial frameworks, it is often necessary to balance both class consistency and feature consistency. Therefore, the weighted sum of results from the two methods at the decision level is adopted as the final prediction, which achieves optimal performance and effectively enhances interpretability.

#### 2.4.2. Conditional Adversarial Domain Adaptation Framework

A conditional domain discriminator (D) is used to supervise the training of the feature extractor (G), directing it toward learning domain-invariant features relevant to both the source and target domains. The adversarial loss ensures effective alignment between the source and target domains:(4)$$L_{adv}\left(\theta_{G},\theta_{D}\right)=-E_{x_{s}∼P_{s}}\left[\log\left(1-D\left(\varphi \left(G\left(x_{s}\right),{\tilde{y}}_{s}\right)\right)\right)\right]-E_{x_{t}∼P_{t}}\left[\log\left(D\left(\varphi \left(G\left(x_{t}\right),{\tilde{y}}_{t}\right)\right)\right)\right]$$

Here, $\theta_{G}$ and $\theta_{D}$ denote the parameters of the feature extractor G and domain discriminator D, respectively. φ represents a multilinear mapping, defined as φ(f,g)=f⊗g, which imposes constraints on the features $G(x_{s})$ and $G(x_{t})$ through the model’s predicted probabilities ${\tilde{y}}_{s}$ and ${\tilde{y}}_{t}$ for the samples, respectively. Detailed information about φ is provided in Ref. [26].

For class classifier (C), it is trained using cross-entropy loss to ensure that it has basic classification ability:(5)$$L_{cls}\left(\theta_{G},\theta_{C}\right)=-E\left(x_{s},y_{s}\right)∼P_{s}\left[y_{s}\log\left(C\left(G\left(x_{s}\right)\right)\right)\right]$$

However, during adversarial training, as the source and target domains need to reduce shift, the discriminative information of disease spectra may be disrupted. This leads to a problem where domains are confused, but the classifier still fails to effectively classify diseases. Therefore, in this study, we introduce a metric constraint (MC) loss based on Fisher linear discriminant analysis [33] into the adversarial learning process, adding an additional constraint during adversarial training. Specifically, suppose $\;G(x_{s}^{u})\;$ is the feature extracted by the feature extractor G from sample $\;x_{s}^{u}\;$, and $\;y_{s}^{u}\;$ is the label of this sample. First, the normalization coefficient $\;T_{m}\;$ is calculated as follows:(6)$$T_{m}=\frac{1}{B_{m}}\sum_{u,v\in \{1,\ldots ,B_{m}\}}{∥G\left(x_{s}^{u}\right)-G\left(x_{s}^{v}\right)∥}_{2}^{2}$$

$\;B_{m}\;$ denotes the number of source domain samples; then, the calculation formula for the MC loss is as follows:(7)$$L_{MC}\left(\theta_{G}\right)=E_{\left(x_{s},y_{s}\right)∼P_{s}}\times \log\frac{\sum_{y_{s}^{u}\neq y_{s}^{v}}\exp(-{∥G(x_{s}^{u}-G\left(x_{s}^{v}\right))∥}_{2}^{2}/T_{m})}{\sum_{y_{s}^{u}=y_{s}^{v}}\exp(-{∥G(x_{s}^{u}-G\left(x_{s}^{v}\right))∥}_{2}^{2}/T_{m})}$$
where $u,v\in \{1,\ldots ,B_{m}\},u\neq v$. The MC loss preserves the structural information of the original data through adversarial training, ensuring the discriminability of representations, thereby improving the performance of adversarial learning.

In summary, the optimization objective of the conditional adversarial domain adaptation part is as follows:(8)$$\max_{\theta_{G},\theta_{C_{m}}}\;\min_{\theta_{D_{m}}}\;\lambda \left(L_{adv}\left(\theta_{G},\theta_{D}\right)-L_{MC}\left(\theta_{G}\right)\right)-L_{cls}\left(\theta_{G},\theta_{D}\right)$$

The overall process of CDAN-PL can be described by pseudo-code as Algorithm 1:

**Algorithm 1** Conditional domain adversarial networks based on voting to generate consistent pseudo labels (CDAN-LP)

**Input:** Source set: $S={\left\{\left(x_{s},y_{s}\right)\right\}}^{N_{s}}$, Target set: $T={\left\{\left(x_{t},y_{t}\right)\right\}}^{N_{t}}$, training epochs E1, batch size: B, parameter: λ, Confidence threshold κ.**Output:** Feature extractor (G), domain discriminators (D), category classifiers (C). Pseudo Label Generator: (P)
1   Initialize ${\left\{D\right\}}^{N}$, ${\left\{C\right\}}^{N}$ with xavier2   **for** e∈1,2,…,E **do**3   Sample a B-batch from $D_{s}={\left\{\left(x_{s},y_{s}\right)\right\}}^{N_{s}}$

$D_{t}={\left\{\left(x_{t}\right)\right\}}^{N_{t}}$

4   Calculate adversarial losses via Equation (Equation 4)5   Calculate source category classification loss via Equation (Equation 5)6   Generate pseudo labels with the Pseudo Label Generator7   Threshold-based filtering of pseudo-labels.8   Calculate MC loss via Equations (6) and (7)9   Calculate CE classification loss via Equation (Equation 2)10  **end for**



## 3. Results

### 3.1. Experimental Setup

For the baseline models, we selected the source-only model (without adaptation) and the classic adversarial domain adaptation model DANN [34]. The version of our framework that does not use pseudo-labels is also employed as a comparison to verify the role of pseudo-labels and their involved operations in the entire model. This model is equivalent to a conditional domain adversarial network (CDAN) constrained by $L_{MC}$, namely CDAN-MC.

The source-only model is obtained by performing supervised training on the source domain data based on the feature extractor and class classifier we adopted. It can represent the performance of our proposed framework on the target domain before migration and intuitively demonstrate the overall effect of our proposed pseudo-label-based conditional domain adaptation framework on domain alignment from the results. The Domain-Adversarial Neural Network (DANN) is a classic adversarial domain adaptation framework, which consists of a feature extractor, a domain discriminator, and a class classifier. The domain discriminator propagates gradients back to the feature extractor through a gradient reversal layer, enabling the latter to learn domain-confused features so as to align the features of the source and target domains. The class classifier outputs category labels based on the features. Ideally, the domain discriminator confuses the feature extractor so that it does not extract domain-related features regardless of whether the input data are from the source or target domain. Meanwhile, the classifier is trained to distinguish the categories of such features, thereby achieving the goal of unsupervised training of the model on the target domain as a whole. In addition, the results of the conditional domain adaptation framework without using pseudo-labels are included as a comparison model, which can reflect the role of operations involving pseudo-labels in feature alignment.

Table 4 and Table 5 present the experimental environment, version information, and model parameter information. All experimental results are the average values of three experiments.

### 3.2. Homologous Disease Migration

To achieve efficient diagnosis of autoimmune diseases, we selected three homologous diseases, namely ankylosing spondylitis (AS), osteoarthritis (OA), and Sjögren’s syndrome (SS), for mutual migration to minimize domain shift. Table 6 shows the experimental results of internal migration among these three diseases. In the migration experiments of homologous diseases, our CDAN-PL achieved the best accuracy, with an average accuracy of 92.3% across six tasks, which was 5.9% and 3.8% higher than that of the source-only model trained on CNN and DANN, respectively.

Analysis of classification results across different diseases, together with comparison to previously published studies, further validates the strong diagnostic capability of our proposed model. When using different diseases as the source domain for cross-disease diagnosis, our framework achieves average performances of 88.1% for AS, 91.6% for OA, and 97.2% for SS. Compared with supervised learning approaches reported in the literature, our method demonstrates comparable or even superior diagnostic performance under unsupervised conditions. For example, Wu et al. [35] employed Fourier transform infrared spectroscopy combined with a multi-scale ResNet, achieving supervised diagnostic accuracies of 88.89% for AS and 88.46% for OA. Similarly, Bhavik Vyas et al. [13] integrated a genetic algorithm with an $SVM_{D}A$ model to analyze salivary Raman spectra from patients with SS, attaining a diagnostic accuracy of 97%. In contrast, our framework achieves comparable or higher performance without the need for labeled data, underscoring its potential value and applicability in real-world diagnostic settings.

An important task of domain adaptation is to align the features of the source domain and the target domain. The t-SNE algorithm [36] is a nonlinear dimensionality reduction algorithm that can reduce high-dimensional data to two or three dimensions, facilitating visualization research. It is used in domain adaptation to easily observe the alignment degree of features between the source domain and the target domain. Figure 3 shows the structural information of the feature distribution space of the source domain and target domain in each group of experiments.

From the feature distributions of the source domain and target domain, it can be observed that the target domain still extracts category-based features even in the case of a complete lack of labels. After dimensionality reduction to a 2D space using the t-SNE algorithm, the features of diseases and healthy controls can be clearly divided into two clusters.

### 3.3. Non-Homologous Disease Migration

Unlike homologous diseases, non-homologous diseases exhibit significant differences in pathogenesis, symptoms, early diagnosis, and disease prognosis. Migration experiments involving non-homologous diseases can effectively verify the generalization ability of the model. We selected cancer as a typical type of non-homologous disease, and esophageal cancer (EC), kidney cancer (KC), and lung cancer (LC)—which have high incidence rates, abundant clinical cases, and easily accessible data and labels—are suitable as source domain data. Table 7 presents the experimental results of migration from the three types of cancer to autoimmune diseases, where CDAN-PL achieved the highest accuracy in all tasks, with an average accuracy of 90.05%, which was 12.85% and 13.65% higher than that of the source-only model and DANN, respectively.

As shown in Figure 4, compared with homologous migration tasks, the category-based features extracted from the target domain data in non-homologous migration tasks are poorer, and most of them do not show clear clustering in the feature distribution. This is also reflected in the classification results: the diagnostic accuracy of non-homologous migration tasks is 2.25% lower on average than that of homologous migration tasks.

## 4. Interpretability

For end-to-end deep network frameworks, Grad-Cam [37] provides an effective method to describe the interpretability of the network. Specifically, Grad-Cam obtains the gradient information of the feature layer through backpropagation on the network’s predicted values, then performs weighted summation and activation on all channels, and finally obtains a set of weights. This set of weights can be regarded as the degree of contribution of different positions in the feature map to the network’s predicted values. We applied this technology to the migration tasks of Raman spectroscopy, enabling the plotting of heatmaps. Figure 5 shows the spectral heatmaps of all homologous disease migration tasks. In Raman spectroscopy, different peak positions reflect specific structures and their vibrational information in the measured sample substances, thereby indicating the presence of different substances. Peak positions with large weights shown in the heatmaps can, to a certain extent, indicate that the substances corresponding to these peaks are of great significance for the diagnosis of autoimmune diseases or may become important biomarkers. To explore this possibility, we performed spectral analysis on the spectra in Figure 5.

Tasks with the same target domain disease are grouped together, dividing the six tasks into three groups, where each group represents the diagnostic task results of OA, AS, and SS, respectively. This study identified peaks with a contribution value greater than 0.5 (shown in red in Figure 5) when different diseases were transferred to the target disease; these peaks were defined as the key characteristic peaks for model-based diagnosis. Table 8 presents the key characteristic peaks across all tasks and their occurrence frequencies. In the six groups of homologous disease transfer tasks, the features identified as key characteristic peaks may represent the common traits shared by homologous autoimmune diseases. These features were effectively expressed and transferred to the target domain during the domain adaptation process. Among them, characteristic peaks such as 924.02 cm^−1^, 929.19 cm^−1^, 990.84 cm^−1^, and 1316.85 cm^−1^ exhibited high contribution in all homologous tasks and were identified as key characteristic peaks, indicating that they may correspond to key biomolecules influencing disease diagnosis. The biomolecules represented by the key characteristic peaks in Table 8 will be analyzed across different diseases to identify specific disease markers.

Proline, valine, phenylalanine, and guanine (B, Z-marker), as well as lipids, contributed more than 0.5 in all tasks, proving, to some extent, their specificity in the diagnosis of autoimmune diseases. Proline is a non-essential amino acid that is essential for the synthesis of collagen, one of the main components of cartilage. It has been shown [38] that circUbqln1 (non-coding RNA) is able to promote OA by affecting proline metabolism; specifically, circUbqln1 upregulates the transcriptional and enzymatic activities of proline dehydrogenase (PRODH), leading to an acceleration of proline deletion, resulting in the accumulation of its metabolite P5C (proline-5-carboxylic acid), which may interfere with the normal collagen synthesis process. Also, proline was one of the best biomarkers for differentiating autoimmune neuroinflammation from controls, with an area under the receiver operating characteristic curve (AUC) of 0.77 [39]. Valine is generally thought to be associated with the development of autoimmune neuroinflammatory disorders, and it is included in a larger group of metabolites that consists primarily of amino acids, amino acid metabolites, and acetyl carnitine. It has also been noted that the serum metabolomics of patients with SS and HC can be distinguished by 21 significant metabolites, including elevated levels of alanine and valine [40]. Methionine or its metabolites may be implicated in the pathogenesis of autoimmune diseases in some cases, e.g., methionine plays a key role in the activity of α1-antitrypsin (AAT), especially in the active center loop of AAT, and oxidation of methionine can affect its inhibition of proteases such as neutrophil elastase, which further affects certain disease processes [41].

Furthermore, in autoimmune diseases such as Sjögren’s syndrome (SS), cholesterol metabolism plays a key role in T-cell biology [42]. Cholesterol helps maintain the stability of cell membranes and regulates their fluidity. In addition, cholesterol is involved in the formation of key structures such as lipid rafts, major histocompatibility complex molecules, and T-cell receptors, all of which are necessary for adaptive immunity.

## 5. Conclusions and Discussion

This study systematically explores the application framework of unsupervised domain adaptation (UDA) technology in Raman spectroscopy analysis, providing an effective solution to address the domain shift problem caused by data discrepancies. The proposed CDAN-PL framework demonstrates significant performance advantages over traditional supervised learning methods in scenarios with extremely limited labeled data and outperforms other state-of-the-art (SOTA) domain adaptation algorithms. Experimental results show that in homologous disease migration tasks, our proposed CDAN-PL achieves the highest accuracy compared to baseline models, with an average accuracy of 92.3% across all tasks. In addition, we selected three common types of cancer as source domain data, which also effectively perform unsupervised diagnosis of autoimmune diseases with an average accuracy of 90.05%, verifying the generalization ability of our model. Furthermore, using Grad-CAM technology, we identified spectral peaks that contribute significantly to diagnostic results, and the substances corresponding to these peaks may become important biomarkers for future diagnosis and research. Overall, this study provides a new perspective and method for the unsupervised diagnosis of autoimmune diseases.

In summary, the CDAN-PL framework has initially verified the feasibility of autoimmune disease diagnosis based on unlabeled Raman spectroscopy. However, this study still has several limitations, including a limited sample size and a single data modality. A total of 297 samples from various autoimmune diseases and their corresponding control groups, as well as 154 samples from various types of cancer and their control groups, were included in this study. Although domain adaptation models typically rely on large-scale data to ensure the reliability of knowledge transfer, we adopted the Synthetic Minority Oversampling Technique (SMOTE) to perform data augmentation separately for the disease group and the control group in each experiment to alleviate the issue of insufficient sample size, increasing the sample size of each category to 300. Nevertheless, considering that excessive use of SMOTE may distort the original statistical distribution of the data [43], no further expansion of the augmentation scale was conducted. In the future, we plan to recruit more real-world samples and develop more specialized algorithms for few-shot domain adaptation, aiming to further improve the performance and application value of Raman spectroscopy in unsupervised and supervised diagnostic tasks.

Furthermore, although the current model relies solely on spectral features derived from blood samples, it is well established that clinical variables such as age and gender can significantly influence both susceptibility to and progression of autoimmune disorders [44,45]. Future extensions of this work could therefore benefit from integrating such demographic and clinical metadata, either through multimodal learning architectures or hybrid predictive frameworks. Incorporating these factors would enable more personalized and context-aware diagnostic support, thereby aligning computational predictions more closely with established clinical practice.

## Figures and Tables

**Figure 1 sensors-25-06186-f001:**
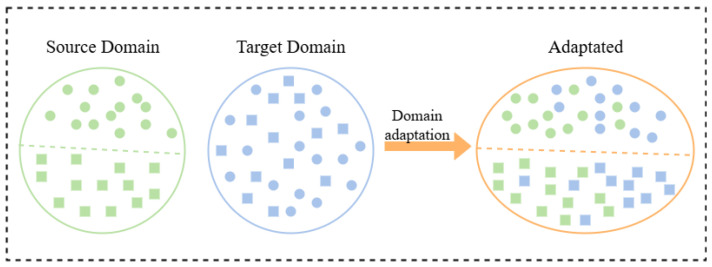
Schematic representation of the domain adaptation.

**Figure 2 sensors-25-06186-f002:**
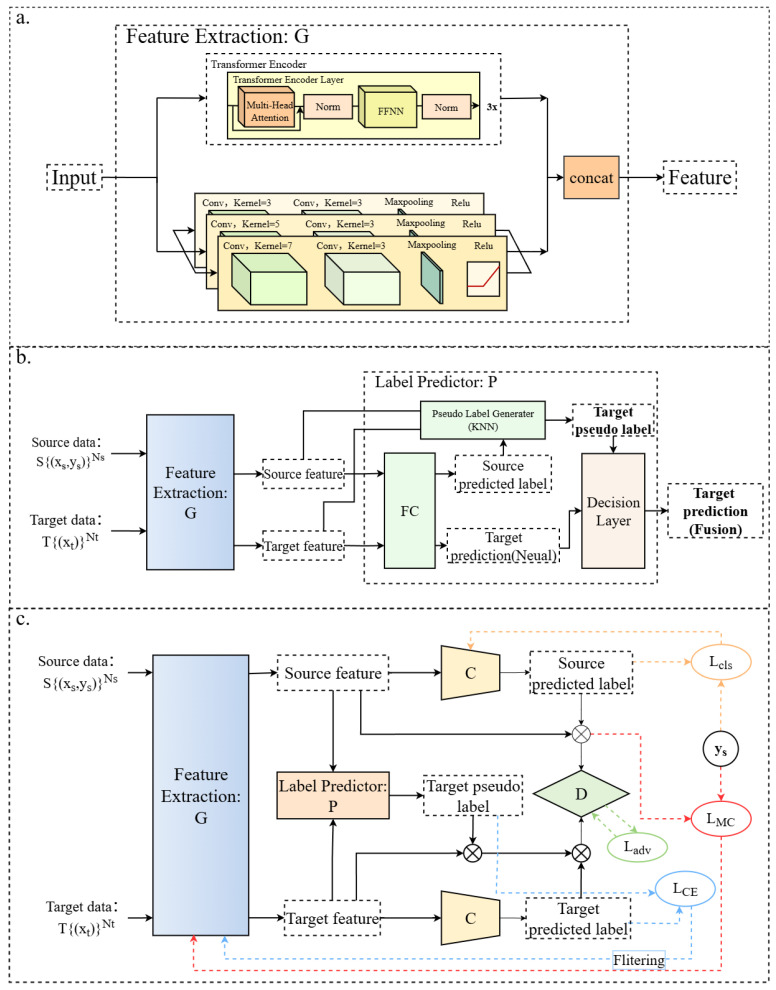
Overall technical framework of CDAN-PL. (**a**). Detailed internal structure of the feature extractor (G). (**b**). Internal structure of the label predictor (P). (**c**). Structural diagram and training process of the CDAN-PL deep model.

**Figure 3 sensors-25-06186-f003:**
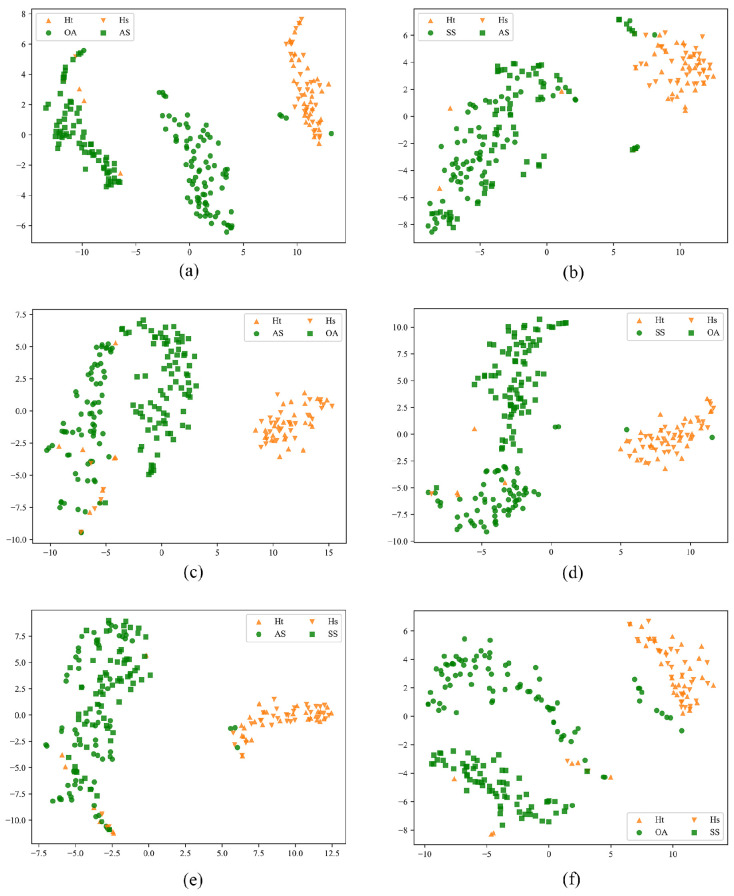
Visualization of the feature distribution for each category of the source and target domains of the six homologous disease migration tasks, respectively. The yellow inverted triangle and yellow triangle are the healthy controls of the source and target domains, respectively, and the green dots are the source domain diseases, and green squares are the target domain diseases: (**a**) Visualization of the feature distribution of OA→AS. (**b**) Visualization of the feature distribution of SS→AS. (**c**) Visualization of the feature distribution of AS→OA. (**d**) Visualization of the feature distribution of SS→OA. (**e**) Visualization of the feature distribution of AS→SS. (**f**) Visualization of the feature distribution of OA→SS.

**Figure 4 sensors-25-06186-f004:**
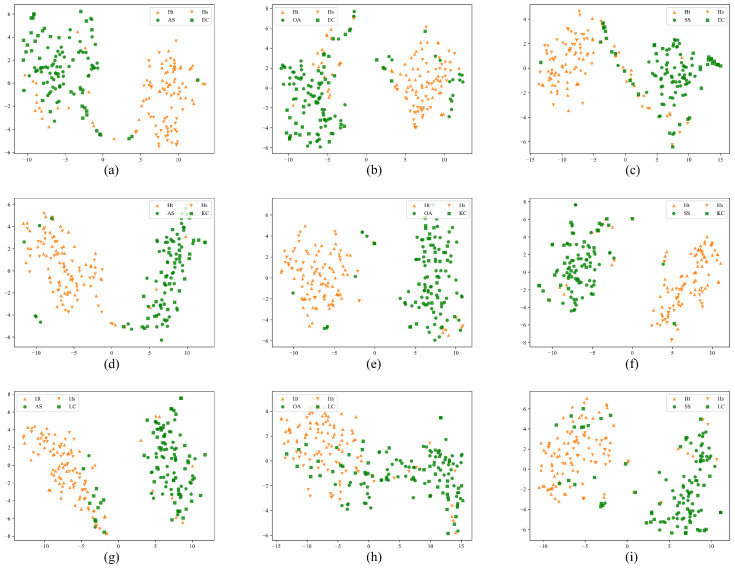
Visualization of feature distribution for each category of source and target domains for the above nine non-homologous disease migration tasks, respectively. Yellow inverted triangles and yellow triangles are the healthy controls of the source and target domains, respectively, while green dots are the source domain diseases, and green squares are the target domain diseases: (**a**) Visualization of the feature distribution of EC→AS. (**b**) Visualization of the feature distribution of EC→OA. (**c**) Visualization of the feature distribution of EC→SS. (**d**) Visualization of the feature distribution of KC→AS. (**e**) Visualization of the feature distribution of KC→OA. (**f**) Visualization of the feature distribution of KC→SS. (**g**) Visualization of the feature distribution of LC→AS. (**h**) Visualization of the feature distribution of LC→OA. (**i**) Visualization of the feature distribution of LC→SS.

**Figure 5 sensors-25-06186-f005:**
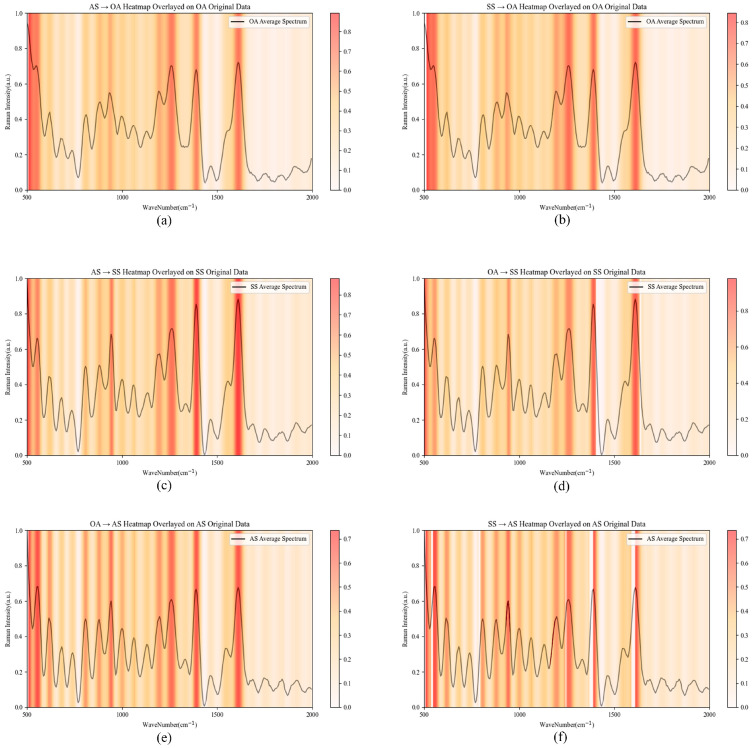
Average spectral heatmap of the homologous disease transfer task. (**a**,**b**) demonstrate the contribution of different Raman spectral peaks in the OA diagnostic task with different diseases as source domains. (**c**,**d**) demonstrate the contribution of different Raman spectral peaks in the AS diagnostic task with different diseases as source domains. (**e**,**f**) demonstrate the contribution of different Raman spectral peaks in the SS diagnostic task for different diseases as source domains.

**Table 1 sensors-25-06186-t001:** Comparison of previous studies on artificial intelligence combined with Raman spectroscopy.

Research	Transfer Learning	Domain Adaptation	Supervised Learning	Unsupervised Learning	Disease Diagnostics
Leng et al. [15]			✓		Cancer
Yong et al. [16]			✓		Osteoarthritis and osteoporosis
Yang et al. [17]			✓		Liver cancer
Cao et al. [18]			✓		Colorectal cancer
Hu et al. [22]	✓			✓	-
Chen et al. [23]	✓			✓	Cancer
Yao et al. [24]	✓	✓		✓	-
Our work	✓	✓		✓	Autoimmune disease

✓: This research conforms to this option.

**Table 2 sensors-25-06186-t002:** Raman spectroscopy sample information.

Disease	Category	Sample Size	Wavenumber Range	Sample Type
Autoimmune disease	Ankylosing Spondylitis	72	500–2000 cm^−1^	Serum
	Osteoarthritis	87		
	Sjögren’s Syndrome	70		
	Healthy Control	68		
Cancer	Esophageal Cancer	37		
	Lung Cancer	44		
	Kidney Cancer	28		
	Healthy Control	45		

**Table 3 sensors-25-06186-t003:** Source and target domain dataset information.

Domain	Dataset	Category	Data Size	SMOTE
Source Domain	Ankylosing Spondylitis Dataset	AS	72	300
		Healthy Control	34	
	Osteoarthritis Dataset	OA	87	
		Healthy Control	34	
	Sjögren’s Syndrome Dataset	SS	70	
		Healthy Control	34	
Target Domain	Ankylosing Spondylitis Dataset	AS	72	
		Healthy Control	68	
	Osteoarthritis Dataset	OA	87	
		Healthy Control	68	
	Sjögren’s Syndrome Dataset	SS	70	
		Healthy Control	68	

**Table 4 sensors-25-06186-t004:** Experimental environment and version information.

	Environment	Version
Language	Python	3.8.0
	MATLAB	R2021a
Dataset division	sklearn.model_selection	
Artificial synthetic dataset generation	torch.DataLoader	
Dataset processing	scipy	1.10.1
Classifier	torch.nn	1.13.0
	torchvision	0.14.0
T-SNE	sklearn.manifold	1.7.1
Calculation	numpy	1.21.2
Data augment	imblearn. over_sampling	
Data normalization	sklearn.preprocessing	1.1.3
GPU	RTX 3090 (24 GB)	

**Table 5 sensors-25-06186-t005:** Model parameter settings.

Model	Parameter
CDAN-LP	Backbone = G, Optimizer = RMSprop, Weight initialization = Xavier, learning rate = 0.00001, Batch = 32, Iteration = 500
Comparison Model	Backbone = ResNet50, Optimizer = SGD, Weight initialization = Xavier, learning rate = 0.00001, Batch = 32, Iteration = 500

**Table 6 sensors-25-06186-t006:** Results of homologous disease transfer experiments.

Source Domain	AS		SS		OA		Average
**Target Domain**	**SS**	**OA**	**AS**	**OA**	**SS**	**AS**	
Source-only	90.62%	83.78%	78.12%	72.97%	90.62%	68.75%	80.81%
DANN	94.3 ± 2.34%	92.8 ± 3.71%	85.6 ± 7.55%	83.8 ± 7.47%	91.8 ± 4.23%	70.3 ± 6.93%	86.4 ± 5.37%
CDAN-MC	93.1 ± 1.25%	90.2 ± 1.32%	81.8 ± 1.25%	86.4 ± 1.71%	94.3 ± 1.25%	85.6 ± 1.53%	88.5 ± 1.38%
CDAN-PL	**99.4 ± 1.24%**	**96.2 ± 2.16%**	**88.1 ± 1.24%**	**87.0 ± 1.08%**	**95.0 ± 1.53%**	**88.1 ± 1.24%**	**92.3 ± 1.41%**

**Table 7 sensors-25-06186-t007:** Results of homologous disease transfer experiments.

Source Domain	EC			KC			LC			Average
**Target Domain**	**SS**	**AS**	**OA**	**SS**	**AS**	**OA**	**SS**	**AS**	**OA**	
Source-only	90.40%	83.30%	72.30%	83.30%	83.30%	80.80%	76.20%	76.20%	48.90%	77.20%
DANN	74.7 ± 1.90%	72.8 ± 1.16%	76.0 ± 3.14%	86.6 ± 3.23%	79.5 ± 1.88%	85.1 ± 1.34%	73.8 ± 3.98%	86.6 ± 1.90%	53.2 ± 1.90%	76.4 ± 2.27%
CDAN-MC	92.3 ± 1.78%	85.7 ± 1.50%	79.1 ± 1.57%	92.8 ± 1.50%	85.2 ± 0.95%	85.1 ± 4.03%	93.8 ± 1.16%	89.5 ± 1.16%	73.2 ± 2.57%	86.3 ± 1.80%
CDAN-LP	**97.1 ± 0.95%**	**85.7 ± 1.50%**	**85.1 ± 1.90%**	**94.7 ± 1.78%**	**87.1 ± 1.17%**	**88.93 ± 2.08%**	**96.2 ± 1.16%**	**91.9 ± 1.16%**	**83.8 ± 1.04%**	**90.05 ± 1.41%**

**Table 8 sensors-25-06186-t008:** Distribution of major peaks and number of occurrences.

Peak (cm^−1^)	Assignment	Occurrence Count
631.18	ν(C—S) gauche (amino acid methionine)	2
845.8	Polysaccharide structure	2
924.02, 929.19	ν(C—C), stretching; probably in amino acids proline and valine (protein band)	6
980.62	C—C stretching β-sheet (proteins)	2
990.84	Single human RBC, phenylalanine, NADH	6
1051.74	C—O stretching, C—N stretching (protein)	2
1249.49	Amide III	2
1254.33	C—N in plane stretching	4
1316.85	Guanine (B, Z-marker)	6
1444.16	Lipid	6
1653.46	Lipid (C=C stretch)	6

## Data Availability

The data are not publicly available due to data privacy regulations. The data presented in this study are available upon request from the corresponding author.

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
