# Peer review of "Unsupervised Domain Adaptation with Raman Spectroscopy for Rapid Autoimmune Disease Diagnosis"

_sensors, 2025, doi:10.3390/s25196186_

Round 1

Reviewer 1 Report

Comments and Suggestions for Authors

The article “Unsupervised Domain Adaptation with Raman Spectroscopy for Rapid Autoimmune Disease Diagnosis” is interesting for its application in the medical field.

These are my suggestions to improve the manuscript.

  1. Introduction part should have more references to strengthen this study (there are plenty of Raman studies who address this subject)
  2. Line 9-11 should be at the end of the introduction part.
  3. Why this study was granted an exemption from informed consent? (line 95-97) In the Institutional Review Board Statement and the Informed Consent Statement, the authors have stated they have obtained the informed consent and the ethics committee approval but in the manuscript they state the study is exempted from informed consent. Please address this issue and provide an explanation for this. 
  4. Chapter 2.2 should be rewritten to be more accurate (2 Raman spectrometers were used?). Why the authors used two different laser wavelengths for different diseases and only 500-2000 cm-1 interval? Useful information can be found also in the interval 2000-4000 cm-1, especially in the case of cancers.

Reviewer 2 Report

Comments and Suggestions for Authors

Unsupervised Domain Adaptation with Raman Spectroscopy for Rapid Autoimmune Disease Diagnosis

This study provides information on Unsupervised Domain Adaptation with Raman Spectroscopy technology in the rapid diagnosis of Autoimmune diseases. This is a good development with the existence of ML models that are able to diagnose complex diseases rapidly and non-invasively such as autoimmune diseases and cancer.

In the introduction, it is not clearly stated what the existing research gaps are and the role of CDAN-PL with UDA in addressing this issue.

It is not clear who made the final diagnosis of autoimmune disease and cancer on samples that used. Is this standard diagnosis based on clinical symptoms and laboratory tests?

The small sample size seems to be a limitation, as adaptive learning requires a significant number of samples for the ML model to achieve high accuracy. Meanwhile, the number of autoimmune diseases is only 229 and cancer is 109.

Autoimmune diseases are typically determined by age and gender. Shouldn't this be a consideration for adapting this ML model? The blood sample used certainly contains thousands of detectable substances, but factors other than the blood sample also need to be considered in the diagnosis. Should be added in discussion.

Reviewer 3 Report

Comments and Suggestions for Authors

The authors contributed to the work titled “Unsupervised Domain Adaptation with Raman Spectroscopy for Rapid Autoimmune Disease Diagnosis.” The manuscript is well-structured and presents the research. It is suitable for acceptance; however, the authors should address several comments before final approval.
1) Please provide a graphical abstract.
2) What are the novelties of unsupervised domain adaptation with Raman spectroscopy? The authors could include a table comparing their investigation with other previously reported studies (for example, artificial intelligence combined with Raman spectroscopy, etc.) regarding benefits, limitations, and other relevant factors. Make sure the conclusions highlight the scientific contribution, applicability, and novelty of the study.
3) The novelty of the work should be highlighted in the introduction.
4) Discuss not only the positive aspects but also the potential challenges and limitations of unsupervised domain adaptation with Raman spectroscopy.
5) Improve the figure resolution, clarify details, and increase the font size of text and labels.
6) Please correct any typos.

Reviewer 4 Report

Comments and Suggestions for Authors

The study discusses the potentials of applying deep learning to ultimately be able to identify spectral data that correspond to autoimmune diseases. There are some questions that need to be addressed and some suggestions that can help improve the quality of the paper:

  1. The Section 3 (Results) is quite short given the scope of the paper and the described Materials and Method in Section 2. Can this section be further expanded? Maybe explain the obtained metrics (accuracies) and how they compare with the recent progress of similar literature. 
  2. The inclusion of spectral heatmap in Figure 5 has made the discussion easier to understand as it supports the previously discussed results of deep learning. It would be better that Figure 5 be further elaborated in relation to Table 8 - what do the occurence values imply and how significant are they in interpreting the results of the spectral data?

Overall, the study presents a substantial approach in processing spectral data and would be significant. Only a few minor edits as mentioned above is needed to improve the manuscript.

Reviewer 5 Report

Comments and Suggestions for Authors

In this paper, the authors propose an approach based on the employment of a conditional domain adversarial network, coupled with Raman spectroscopy for diagnostic purposes. In particular, the authors focused their attention on the diagnosis of autoimmune diseases. In addition, they adopted a domain adaptation approach to realize pseudo-labels, with the aim of addressing the problem of missing data. The Raman spectra were obtained from serum samples. The manuscript is complete and well-written, and it falls within the scope of the journal. For this reason, my feedback about the manuscript is positive. In the following, I’ve attached some minor indications to further improve the quality of the paper:
1. Row 106: since in other parts of the subsection 2.2 the authors adopted the passive voice, I suggest using the same conjugation. In particular “transfer 10μL of serum sample...” should be “10μL of serum sample was transferred...”. Same problem at rows 108 (“Collect spectra...”) and 113 (“Randomly select...”);
2. Why did you employ two different wavelengths for autoimmune diseases
and cancer samples?
3. Row 127: I don’t fully agree with the statement ‘’which do not carry chemical information of the measured sample”, specifically for the background radiation. The baseline signal component contains information about autofluorescence, which is representative of the molecular nature of
the sample. Rather, in this case, the baseline removal is justified by the interference of this signal component with the Raman signal.
4. Row 136: the authors should specify the type of normalization adopted;

5. Row 146: While referring to SMOTE, the authors state “while preserving the fundamental characteristics of the original class”. I don’t fully agree with this statement. Especially when the number of the generated examples is large, SMOTE could significantly affect the statistical distribution. According to what is written in rows 149-150, the number of generated examples was high. I suggest at least highlighting this aspect in the Conclusions section, and addressing the problem of monitoring the impact of SMOTE on the statistical distribution in future works.
6. subsection 2.4: although it could be obvious, I suggest specifying the meaning of xS , yS , xt, yt to enhance the readability;
7. By observing the diagram in Fig. 2 (a), it seems that G is the whole algorithm, while in rows 173-174, G is described as a part of the dual-branch structure. Please clarify.
